# Binge alcohol drinking before pregnancy is closely associated with the development of macrosomia: Korean pregnancy registry cohort

**Seul Koo[1]☯, Ji Yeon Kim[1]☯, Ji Hye Park[1], Gu Seob Roh[2], Nam Kyoo Lim[1], Hyun Young Park[1], Won-Ho Kim**📧[1] *

**1** Division of Cardiovascular Disease Research, Department of Chronic Disease Converengence Research, Korea National Institute of Health, Cheongju, Chungbuk, Republic of Korea, **2** Department of Anatomy and Neurobiology, Gyeongsang National University, Jinju, Gyeongnam, Republic of Korea

☯ These authors contributed equally to this work.
* jhkwh@nih.go.kr

**Data Availability Statement:** The data sets analyzed during the current study are not publicly available due to current Korean law regulating

## Abstract

### Background

Alcohol drinking during pregnancy has been well-known to cause the detrimental effects on fetal development; however, the adverse effects of pre-pregnancy drinking are largely unknown. We investigate whether alcohol drinking status before pregnancy is associated with the risk for macrosomia, an offspring's adverse outcome, in a Korean pregnancy registry cohort (n = 4,542) enrolled between 2013 and 2017.

### Methods

Binge drinking was defined as consuming ≥5 drinks on one occasion and ≥2 times a week, and a total 2,886 pregnant, included in the final statistical analysis, were divided into 3 groups: never, non-binge, and binge drinking.

### Results

The prevalence of macrosomia was higher in binge drinking before pregnancy than those with never or non-binge drinking (7.5% vs. 3.2% or 2.9%, $p = 0.002$). Multivariable logistic regression analysis demonstrated an independent association between macrosomia and prepregnancy binge drinking after adjusting for other confounders (adjusted odds ratio = 2.29; 95% CI, 1.08–4.86; $p = 0.031$). The model added binge drinking before pregnancy led to improvement of 10.6% (95% CI, 2.03–19.07; $p = 0.0006$) in discrimination from traditional risk prediction models.

### Conclusion

Together, binge drinking before pregnancy might be an independent risk factor for developing macrosomia. Intensified intervention for drinking alcohol in women who are planning a pregnancy is important and may help prevent macrosomia.

release of personal information, even in the cases of research purposes. All data are currently only available to the researchers participated in the establishment of Korean pregnancy registry study; however, data analysis collaborations may be possible through specific research proposals. And also, the datasets used and analyzed during the current study are available from the corresponding author on reasonable request and requested data may be disclosed only after approval from the Institutional Review Board (IRB) of Cheil General Hospital and CHA Gangnam, Medical Center, Seoul, Korea. Further information can be requested by e mailing the principal investigator (hmryu2012@naver.com). Meanwhile, you can also request a dataset from Dr. Lee KH (khlee3789@korea.kr), Division of Genetic Epidemiology, Korea National Institution of Health, which supports and operates this research cohort project.

**Funding:** - WHK & HYP - WHK & HYP 4800-4861-303 and 4800-4861-312, 2012-NG63002-00, 2017-NI63005-00 - Intramural research grant from the Korean National Institute of Health - WHK: Manager for 2nd Research project and study design, data collection and analysis, decision to publish, writing the manuscript, supervision and edit review - HYP: General manager of Research project for Women Health

**Competing interests:** The authors have declared that no competing interests exist.

**Abbreviations:** GDM, gestational diabetes mellitus; NRI, net reclassification improvement; AUROCs, the area under the receiver operating characteristic curves; Hb, haemoglobin; Hct, hematocrit; Plt, Platelets; WBC, white blood cells; AST, aspartate aminotransferase; ALT, alanine aminotransferase; BUN, blood urea nitrogen.

## Introduction

Maternal alcohol drinking during pregnancy can have negative outcomes for both mother and infant [1, 2]. A lot of evidence regarding the harmful effects of alcohol drinking during pregnancy on maternal and prenatal health risks has been continuously accumulating, while little is yet known about the exact influence of maternal drinking before pregnancy on fetal development and growth. Although reported rates of alcohol use among young women vary depending on the differences in cultural factors each country, more than 50% of young women in many countries drink alcohol before pregnancy [3, 4]. Despite the recommendations of alcohol abstinence for women who are pregnant or planning a pregnancy [5, 6], previous systematic review and meta-analysis studies demonstrated that alcohol use during pregnancy is common in many countries; especially, Russia (36.5%, the United Kingdom (41.3%), Denmark (45.8%), Belarus (46.6%) and Ireland (60.4%) showed the highest rates [7]. In the United States, approximately 10% of pregnant women admit to alcohol drinking in the past month, and about 50% of them still admit to drinking at some point during their first trimester, often prior to being aware that they are pregnant [8], and in Australia, where pregnant women are recommended not to drink alcohol at any point during pregnancy, 72% did so [9]. In fact, about 40~60% of pregnancies in each country are unplanned, even well-informed and compliant women may have unwittingly consumed alcohol in pregnancy [8, 10–12]. Consequently, alcohol drinking before pregnancy may be closely associated with unintended fetal alcohol exposure, and it may be a causal factor for detrimental maternal and fetal health. Several studies demonstrated that alcohol is teratogenic and fetotoxic, and passes freely across the placenta to the unborn baby at levels at least equal to that of the mother [13]. It is also widely acknowledged that prenatal alcohol exposure can have a negative impact on growth before and after birth, miscarriage, stillbirth, and preterm birth [14, 15]. However, up to 60% of women who drink alcohol in the preconception period do not recognize pregnancy until the fourth to sixth week of gestation [14]. In fact, it is well known that early stage of pregnancy is an important period to prepare for homeostasis of energy metabolism required for fetal development or growth [16]. In particular, since most major organs, such as the limbs, eyes, and ears, start to form and develop at an early stage [17], acute or chronic alcohol consumption before pregnancy or during early pregnancy may alter the first adjustment of fetal development and growth and may trigger the teratogenic effects of alcohol consumption-mediated fetal developmental disorders. However, there remain significant challenges for the patterns or intensity of alcohol drinking in pregnant women. Furthermore, there is little consensus worldwide concerning the effects of light or moderate maternal drinking before and during pregnancy on fetal development and offspring growth. Some guidelines recommend that women abstain completely drinking alcohol from attempting to get conception until after pregnancy [18], whereas others recommend that women are allowed to drink one or two units 2–3 times per week during pregnancy [19]. The lack of consensus is because there is little study to analyze the outcomes for important confounders of fetal development and offspring growth, such as maternal smoking, physical activity, and body mass index, or for the status of alcohol drinking before and during pregnancy. Most previous studies have been focused on the detrimental effects of alcohol drinking during pregnancy; and thus it is recommended to stop alcohol drinking during pregnancy for the preventing fetal complications [20, 21]. However, the effects of ethanol consumption before pregnancy on the progressive development of the fetus and postnatal growth remain obscure. Moreover, despite the lack of evidence on the effects of alcohol drinking before pregnancy on the progressive development of the fetus and postnatal growth, some guidelines remain state that consumption of small amounts before pregnancy

recognition is unlikely to be a risk to the unborn baby [22, 23]. Although not clinical results for women of childbearing age, we provided direct evidence that in mice, ethanol consumption before pregnancy is closely associated with the abnormal development of the pup, including macrosomia and growth retardation, which are correlated with maternal metabolic disorders [17]. Therefore, to reduce the risk of adverse birth outcomes, clinical evidence of the association between pre-pregnancy alcohol intake and fetal complications is required.

Birth weight of infant, which is a significant fetal outcome, is well-known as a major determinant of an infant's immediate and child future health [24, 25]. The prevalence of infant macrosomia, defined as a birth weight greater than 4000 g [26], has been a rising over the last two to three decades in different countries across the world [27–30]. Also, macrosomia predisposes newborns to altered growth development and increases the risk of obesity in childhood and associated co-morbidities, such as hypertension, cardiovascular disease, and cancer, later in life [31]. Indeed, pregnancies with macrosomia are associated with serious maternal adverse outcomes such as cesarean section, prolonged labor, postpartum hemorrhage, and obstetric anal sphincter injury, resulted in significant increase of public health problems and its related-socioeconomic costs in both mother and fetus [32–34]. Therefore, it is important to identify the detrimental risk factors and prevent an increased risk of macrosomia. In this study, we clinically investigated whether maternal alcohol drinking status before pregnancy is associated with an increased risk of macrosomia and whether binge-alcohol drinking before pregnancy may be an independent risk factor for incident macrosomia, using the database from the Korean Pregnancy Registry cohort conducted by the Korea National Institute of Health (KNIH).

## Methods

### Study participants

Korean Pregnancy Registry Cohort was established in 2013 to investigate the prevalence and risk factors of pregnancy complications among Korean pregnant women and it is the only one established with support from Korea National Institute of Health, Korea Disease Control and Prevention Agency (KNIH-KDCA). Between March 2013 to January 2017, all pregnant women who visited Cheil General Hospital and CHA Hospital for antenatal care during the first trimester were asked to participate in the study. These two hospitals are representative obstetrics and gynecology hospitals located in Seoul, the capital of South Korea, and approximately 5,000 and 2,500 deliveries take place at these facilities per year, respectively. Trained research nurses explained the study in detail and obtained written informed consent from participants and assisted them in completing the interview-based questionnaires. The approved pregnant women were enrolled at first trimester (around 8–13 weeks' gestation) and re-visited during the following their gestation period at second trimester (around 24–28 weeks' gestation), third trimester (around 36–40 weeks' gestation), delivery, and postpartum, respectively. In this ongoing study, a total of 4,542 pregnant women were recruited between March 2013 and May 2017. Of those, we initially selected 3,472 pregnancies with singleton and complete follow-up data (include those with valid (non-missing) data on both binge drinking and macrosomia and exclude those with follow-up loss (n = 1,021) and multiple pregnancy (n = 49)). Of the remaining 3,472 subjects, to minimize heterogeneity ("noise"), which can mask the effect of certain factors or intervention, subjects who had been diagnosed before pregnancy with the following diseases were excluded: (i) pre-gestational diabetes mellitus, (ii) hypertension, (iii) hyperthyroidism, (iv) congenital heart disease, (v) chronic kidney disease, (vi) asthma or atopic dermatitis, (vii) autoimmune disease, (viii) hepatitis A, hepatitis B, or hepatitis C, (viiii) depression, (x) epilepsy, (xi) tuberculosis and (xii) polycystic ovary

syndrome. Finally, a total 2,886 pregnant women were included in the final statistical analysis and categorized according to alcohol-drinking status (S1 Fig).

## Categorization of alcohol-drinking status before pregnancy

Alcohol-drinking status was examined in three occasions: 'Never drinking', 'Former drinker (stopped during pregnancy or before pregnancy)' and 'Current drinker' at the visit of the first trimester. If women answered 'Former drinker' or 'Current drinker', they were then further asked average frequency of alcohol-drinking (≤1/month, 1-2/month, 2-3/week, ≥4/week, everyday) and quantity of alcohol-drinking per drinking day (1–2 drinks, 3–4 drinks, 5–6 drinks, 7–9 drinks, ≥10 drinks) in their lifetime before pregnancy. In these questions, a 'drinks' means the standard cup for each type of drink. In this study, we considered that a standard drink contains 12 grams of pure alcohol in any drink [35]. Among ever drinker (former or current), the number of drinks (cup) consumed per month was calculated by using frequency of alcohol-drinking and quantity of alcohol-drinking per drinking day before pregnancy. And also, the number of drinks (cups) was re-calculated into the ounce (oz), a unit of weight equal to approximately 28.349 grams. Firstly, the participants were categorized according to the number of drinks (cup or ounce) consumed per month as never drinking, ≤10 cup (4.2 oz), 10 cup (4.2 oz)< - ≤20 cup (8.5 oz), 20 cup (8.5 oz)< - ≤ 30 cup (12.7 oz), and >30 cup (12.7 oz). In addition, with regard to risk of binge drinking, binge drinking was defined as consuming ≥5 drinks on one occasion and ≥2 times a week [36, 37], and subjects were divided into 3 groups: never drinking (n = 561, 19.4%), non-binge drinking (n = 2,099, 72.7%), and binge drinking (n = 226, 7.8%). And also, women (n = 2,325) with 'Ever drinker' at the first trimester were furthered asked when they stopped drinking. Among ever drinker, the majority of women (85.6%, 1,990/2,325) stopped to drink alcohol before pregnancy and the remaining 14.3% (332/2,325) of the women stopped within the first trimester to recognize the pregnancy. At the time of enrollment in the first trimester, only three women (0.1%, 3/2,325) were drinking. Most of women (99.9%) with 'Ever drinker' stopped drinking at an early time point before pregnancy and to recognize the pregnancy. Nevertheless, to exclude the harmful effects of alcohol drinking within the first trimester, the results were compared with groups except for 332 women who drank during the first trimester. As a result analyzing the group including or excluding 332 women in 'Ever drinker', there was no significant difference in the prevalence of macrosomia and the overall results.

## Definition of macrosomia and its risk factors

Macrosomia was defined by an offspring birth weight more than 4,000g, which has been proposed in previous studies [17, 19]. Stratified analysis using the severity of maternal alcohol drinking status allowed for the prediction of risk of developing macrosomia in women with or without high-risk of traditional risk factors for macrosomia [38]. Traditional risk factors were classified according to following as: Maternal age (<35, low-risk; ≥35, high-risk), prepregnancy BMI (<25, low-risk; ≥25, high-risk), parity (nulliparous, low-risk; ≥1, high-risk), GDM (No, low-risk; Yes, high-risk). There is no subject with GDM, who has binge drinking status (n = 0). GDM was diagnosed using a two-step method as described in previous study [39]. Briefly, universal screening with a 50-g glucose challenge test (GCT) was conducted between 24 and 28 weeks. If the result of the GCT was 140 mg/dL or more, a confirmatory oral glucose tolerance test (OGTT) was performed. The Cheil General Hospital used the 75-g OGTT as the confirmatory test and used the new diagnostic criteria from the International Association of Diabetes and Pregnancy Study Group (at least on abnormal value: fasting glucose ≥92 mg/dL, 1-hour glucose ≥180 mg/dL, or 2-hour glucose ≥153 mg/dL). The CHA Gangnam Medical Center used the 100-g OGTT and the Carpenter-Coustan criteria for GDM diagnosis (2 or

more abnormal value: fasting glucose ≥95 mg/dL, 1-hour glucose ≥180 mg/dL, 2-hour glucose ≥155 mg/dL, or 3-hour glucose ≥140 mg/dL). The participants who were diagnosed GDM were sent to endocrinologist and management methods were determined after clinical evaluation. In addition, modifiable lifestyle factors were classified according to following as: Smoking before and during pregnancy (Former or current, low-risk; None, high-risk) [40–42] and Physical activity before and during pregnancy (More than moderate, low-risk; None or light, high-risk); and analyzed their effects on the risk of developing macrosomia by maternal alcohol drinking status. Physical activity before and during pregnancy was queried at the first antenated visit and further asked at each visit for frequency and duration of walking, moderate and vigorous-intensity activity [39]. Based on these questionnaires, physical activity levels were categorized according to the following criteria: None or light in physical activity (almost sedentary lifestyle, office work, and a housewife with few housework, etc.) and More than moderate in physical activity (manufacturing, architect, farmer, athlete, housewife with lots of housework, etc.). Also, congenital anomalies are diagnosed before and after birth through imaging methods, such as ultrasound or fetal magnetic resonance imaging (MRI) [43]. Ultrasound examinations were performed at approximately 20 weeks of gestational age. An additional level II ultrasound examination or fetal magnetic resonance imaging (MRI) was performed if fetal anomalies were found on ultrasound examination. Especially, before the fetal MRI was performed, all participants were informed in writing or orally about the safety of the technique and the process and method of the procedure, and the fetal MRI was performed only when they understood and gave their consent. The radiologist was provided with information on the clinical history and the findings of the detailed ultrasound. The results of the ultrasound and fetal MRI were discussed by the specialized radiologist, neonatologists, and obstetricians of the Cheil and CHA hospitals. Complementary invasive tests such as aminocentesis and chronic villus sampling (CVS), which are highly sensitive and specific for the diagnosis of chromosomal or genetic disorders of the fetus and infection risk, are also performed for accurate diagnosis [43]. As well, maternal blood can be used to screen for placental markers to aid in prediction of risk of chromosomal abnormalities or genetic defects [43].

## Statistical analysis

Categorical variables were evaluated using the Fisher's exact test or chi-square test with Cochran-Armitage test for trend and expressed as number (n, %). An analysis of variance (ANOVA) with Tukey's test for post hoc comparisons was conducted for continuous variables, and continuous variables were presented means ± SD. In subanalysis, pearson's correlation analyses was used to investigate the linear association between the number of drinks (cup or oz) per month and newborn's birth weight. Based on the guidelines presented at Mayo Clinical Center, we used a multivariable logistic regression model to assess whether maternal alcohol-drinking was associated with the risk of macrosomia independently of potential confounders and traditional risk factors for macrosomia. We estimated odds ratios (OR) and 95% confidence interval (95% CI) while adjusted for confounder. The multivariable models were used as follows as: Model 1 included adjustment for demographic factors; maternal age (year), education level (high school or less/college/graduate school or more) and monthly income (low, mid-low, mid-high or high). Model 2 included Model 1 adjustment plus lifestyle factors; smoking (none/former or current) and physical activity (none or light/more than moderate). Model 3 included Model 2 adjustment plus traditional risk factors for macrosomia; gestational age (weeks), pre-pregnancy body mass index (BMI) (kg/m$^2$), parity (the number of deliveries), newborn's gender and gestational diabetes (yes/no). In addition, we tested the discrimination and reclassification as accuracy measurement of the developed models using the cohort. To

compare the discrimination ability of the models, the area under the receiver operating characteristic curves (AUROCs) were obtained. The statistical difference between the AUROC for the two models was tested using the method of DeLong *et al.* [44]. The user-category net reclassification improvement (NRI) [45] was calculated to evaluate improvements in the reclassification for both risk models with and without the binge drinking before pregnancy. All statistical analyses were performed by using SAS version 9.4 (SAS Institute Inc, Cary, NC), and a P-value $< 0.05$ was considered as statistically significant.

## Study approval

All Participants provided written informed consent, and the study protocol was approved by the Institutional Review Board (IRB) of Cheil General Hospital (IRB number: CGH-IRB-2013-10), CHA Gangnam, Medical Center (IRB number: 2013-14-KNC12-018), and the Korea Centers for Disease Control and Prevention (IRB No. 2013-10EXP-02-P-E) separately. It was emphasized to all participants to all participants that they were free to withdraw from any part of the study at any point in time.

## Results

### Maternal demographic and biochemical characteristics

To reinforce the evidence regarding the dangers of maternal drinking before conception, 2,886 pregnancy women who were finally included in the Korean pregnancy-registry cohort were classified according to maternal alcohol-drinking status before pregnancy into three groups as being never drinking (n = 561, 19.4%), non-binge drinking (n = 2,099, 72.7%), and binge drinking (n = 226, 7.8%), respectively (Table 1). A mean age of all participants was 33.2 ± 3.7 (range 20–45 years) and there was no significant difference depending on alcohol-drinking status. The prepregnancy BMI of binge drinking group was higher than never drinking group (21.0 ± 3.0 vs. 21.6 ± 3.0 kg/m$^2$, $p < 0.05$). Compared with never and non-binge drinking subjects, subjects with binge drinking status had lower levels of education and household income, and had an 'other' marital status.

We also found that women with binge drinking pattern were more likely to be smoking, high exercise, and primiparous than those with never or non-binge drinking. On the other hand, since all subjects participated in this study were Korean, the racial comparison results could not presented. In laboratory test (Table 2), the levels of total cholesterol at the first trimester were significantly elevated in binge drinking groups compared to never drinking groups, whereas creatinine and albumin levels were decreased in binge drinking groups. Also, total protein levels were significantly decreased in binge drinking groups compared to non-binge drinking groups. In result of third trimester, the change in creatinine levels was the same as those of the first trimester, whereas the levels of total protein and albumin, which were decreased in the first trimester, were reversely increased in binge drinking groups compared to never or non-binge drinking groups. Interestingly, the levels of total cholesterol, which were elevated in the first trimester, were not significantly changed in the third trimester. Interestingly, significant increases of hemoglobin, hematocrit, and ALT levels were newly observed in binge drinking groups of the third trimester compared to non-binge drinking groups.

### Maternal binge drinking before pregnancy is associated with the changes of obstetric and offspring outcomes

Next, we examined the influence of alcohol-drinking status before pregnancy on obstetric outcomes (Table 3). Similar to the previous study, at the first trimester, subjects with binge

**Table 1. Demographic characteristics according to maternal alcohol-drinking status before pregnancy.**

| | All participants (n = 2,886) | Never drinking (n = 561) | Ever drinker[†] | | p-value |
|---|---|---|---|---|---|
| | | | Non-binge drinking (n = 2,099) | Binge drinking (n = 226) | |
| Maternal age (year) | 33.2 ± 3.7 | 33.2 ± 3.8 | 33.3 ± 3.7 | 32.9 ± 4.0 | 0.352 |
| Maternal age | | | | | |
| ≤29 | 487(16.9) | 85 (15.2) | 357 (17.0) | 45 (19.9) | **0.483** |
| 30–34 | 1352 (46.8) | 261 (46.5) | 982 (46.8) | 109 (48.2) | |
| 35–39 | 883 (30.6) | 182 (32.4) | 642 (30.6) | 59 (26.1) | |
| ≥40 | 164 (5.7) | 33 (5.9) | 118 (5.6) | 13 (5.8) | |
| Maternal Pre-pregnancy BMI (kg/m$^2$) | 21.1 ± 2.9 | 21.0 ± 3.0[a] | 21.1 ± 2.9[ab] | 21.6 ± 3.0[b] | **0.021** |
| Education | | | | | |
| High school or less | 238 (8.3) | 46 (8.2) | 159 (7.6) | 33 (14.6) | **0.0003** |
| College | 2156 (74.7) | 411 (73.3) | 1573 (74.9) | 172 (76.1) | |
| Graduate school or more | 492 (17.1) | 104 (18.5) | 367 (17.5) | 21 (9.3) | |
| Monthly income (KRW) | | | | | |
| Low (<3 million) | 361 (12.5) | 71 (12.7) | 238 (11.3) | 52 (23.0) | **< .0001** |
| Mid-low (3–4 million) | 496 (17.2) | 114 (20.3) | 344 (16.4) | 38 (16.8) | |
| Mid-high (4–5 million) | 623 (21.6) | 123 (21.9) | 457 (21.8) | 43 (19.0) | |
| High (>5 million) | 1406 (48.7) | 253 (45.1) | 1060 (50.5) | 93 (41.2) | |
| Marital status | | | | | |
| Currently married | 2799 (97) | 550 (98) | 2041 (97.2) | 208 (92) | **< .0001** |
| Other[‡] | 87 (3.0) | 11 (2.0) | 58 (2.8) | 18 (8.0) | |
| Smoking | | | | | |
| None | 2594 (89.9) | 532 (94.8) | 1910 (91.0) | 152 (67.3) | **< .0001** |
| Former or current | 292 (10.1) | 29 (5.2) | 189 (9.0) | 74 (32.7) | |
| Physical activity | | | | | |
| None or light[§] | 1089 (38.9) | 236 (45.0) | 776 (37.8) | 77 (34.5) | **0.004** |
| More than moderate[§§] | 1712 (61.1) | 288 (55) | 1278 (62.2) | 146 (65.5) | |
| Parity (the number of deliveries) | | | | | |
| Nulliparous | 1723 (59.7) | 333 (59.4) | 1225 (58.4) | 165 (73) | **0.001** |
| 1 | 1013 (35.1) | 197 (35.1) | 766 (36.5) | 50 (22.1) | |
| 2 or more | 150 (5.2) | 31 (5.5) | 108 (5.2) | 11 (4.9) | |

Data are expressed as mean ± standard deviation (SD) or n (%). The p-value is a comparison between the three groups. Bold values are statistically significant findings (p<0.05).

[a,b] Different letters represent statistical difference by Tukey's multiple comparison test.

[†] Ever drinker included former (n = 2,322) and current drinker (n = 3).

[‡] Included never-married/cohabit/separated/divorced/widowed.

[§] None or light in physical activity: almost sedentary lifestyle, office work, and a housewife with few housework, etc.

[§§] More than moderate in physical activity: manufacturing, architect, farmer, athlete, housewife with lots of housework, etc.

BMI, body mass index. KRW, Korean Won.

drinking exhibited a marked difference in perinatal depression compared to those with never or non-binge drinking (25.6% vs. 18.6% or 18.1%, respectively; $p = 0.026$), whereas there were no significant differences in the second trimester. However, the prevalence of depression has significantly risen again in women with binge drinking in the visit of both the third trimester (16.8% vs. 9.9% or 13.6%, respectively; $p = 0.036$) and postpartum (26.5% vs. 14.6% or 14.5%, respectively; $p = 0.000$). The other obstetric outcomes depending on alcohol-drinking status were not significant difference.

**Table 2. Clinical characteristics according to maternal alcohol-drinking status before pregnancy.**

| | All participants (n = 2,886) | Never drinking (n = 561) | Ever drinker[†] | | p-value |
|---|---|---|---|---|---|
| | | | Non-binge drinking (n = 2,099) | Binge drinking (n = 226) | |
| Measured at 1st trimester[‡] | | | | | |
| Hb (g/dL) | 12.7 ± 0.9 | 12.7 ± 0.9 | 12.6 ± 0.9 | 12.6 ± 0.9 | 0.554 |
| Hct (%) | 37 ± 2.6 | 37.2 ± 2.6 | 37 ± 2.6 | 36.9 ± 2.6 | 0.123 |
| Plt (x10³/uL) | 246.9 ± 51.6 | 245.1 ± 48.8 | 246.7 ± 52.2 | 253.1 ± 52 | 0.153 |
| WBC (x10³/uL) | 8.16 ± 1.98 | 8.11 ± 1.96 | 8.18 ± 2 | 8.1 ± 1.86 | 0.668 |
| FBG (mg/dL) | 84.5 ± 12.2 | 85.4 ± 14.1 | 84.4 ± 12 | 83.8 ± 9 | 0.183 |
| AST (IU/L) | 18.2 ± 7.2 | 18.4 ± 10.7 | 18.1 ± 6.1 | 18.4 ± 4.9 | 0.536 |
| ALT (IU/L) | 13.4 ± 12.8 | 14.5 ± 23.9[a] | 13.0 ± 8.0[b] | 14.3 ± 8.9[ab] | **0.027** |
| BUN (mg/dL) | 8.08 ± 2.09 | 8.26 ± 2.28 | 8.03 ± 2.03 | 8.1 ± 2.13 | 0.079 |
| Creatinine (mg/dL) | 0.55 ± 0.12 | 0.57 ± 0.14[b] | 0.55 ± 0.12[a] | 0.53 ± 0.11[a] | **0.001** |
| Total protein (g/dL) | 6.92 ± 0.40 | 6.91 ± 0.40[ab] | 6.93 ± 0.39[b] | 6.85 ± 0.43[a] | **0.023** |
| Albumin (g/dL) | 4.17 ± 0.26 | 4.20 ± 0.26[b] | 4.17 ± 0.25[ab] | 4.13 ± 0.26[a] | **0.006** |
| Total cholesterol (mg/dL) | 175.6 ± 28.4 | 172.5 ± 30.1[a] | 176.1 ± 28.0[b] | 178.3 ± 27.8[b] | **0.015** |
| Measured at 3rd trimester[‡] | | | | | |
| Hb (g/dL) | 12.3 ± 1.0 | 12.3 ± 1.0[ab] | 12.3 ± 1.0[a] | 12.5 ± 1.0[b] | **0.027** |
| Hct (%) | 36 ± 2.8 | 36.0 ± 2.8[ab] | 35.9 ± 2.8[a] | 36.5 ± 2.9[b] | **0.030** |
| Plt (x10³/uL) | 217.1 ± 51 | 218.2 ± 48.3 | 216.8 ± 51.5 | 217.6 ± 52.9 | 0.836 |
| WBC (x10³/uL) | 8.75 ± 2.05 | 8.73 ± 1.98 | 8.73 ± 2.09 | 8.96 ± 1.9 | 0.292 |
| FBG (mg/dL) | 81.1 ± 13.1 | 82.2 ± 14.7 | 80.7 ± 12.5 | 82.2 ± 14.7 | **0.037** |
| AST (IU/L) | 20.6 ± 7.7 | 20.9 ± 10.9 | 20.5 ± 6.8 | 20.0 ± 4.7 | 0.381 |
| ALT (IU/L) | 12.7 ± 12.4 | 12.7 ± 15.7[ab] | 12.4 ± 8.0[b] | 14.9 ± 28.1[a] | **0.021** |
| BUN (mg/dL) | 7.88 ± 2.1 | 8.01 ± 2.11 | 7.83 ± 2.1 | 8.00 ± 2.15 | 0.147 |
| Creatinine (mg/dL) | 0.54 ± 0.12 | 0.55 ± 0.13[b] | 0.53 ± 0.11[a] | 0.52 ± 0.10[a] | **0.0003** |
| Total protein (g/dL) | 6.25 ± 0.37 | 6.24 ± 0.36[a] | 6.25 ± 0.38[a] | 6.32 ± 0.35[b] | **0.018** |
| Albumin (g/dL) | 3.62 ± 0.19 | 3.60 ± 0.2[a] | 3.62 ± 0.19[b] | 3.66 ± 0.18[c] | **0.0003** |
| Total cholesterol (mg/dL) | 267.2 ± 44.3 | 268.3 ± 44.7 | 267.5 ± 44.3 | 262.6 ± 43.2 | 0.252 |

Data are expressed as mean ± SD. The p-value is a comparison between the three groups. Bold values are statistically significant findings (p<0.05).

[a,b]Different letters represent statistical difference by Tukey's multiple comparison test.

[†]Ever drinker included former (n = 2,322) and current drinker (n = 3).

[‡]The first and third trimester means around 8–13 and 36–40 weeks, respectively. Hb, haemoglobin; Hct, hematocrit; Plt, Platelets; WBC, white blood cells; FBG, fasting blood glucose; AST, aspartate aminotransferase; ALT, alanine aminotransferase; BUN, blood urea nitrogen.

Additionally, the relationships between maternal alcohol drinking before pregnancy and offspring's outcomes are exhibited in Table 4. The offspring groups from women with binge drinking had greater birth weight compared with never or non-binge drinking groups (3,322.7 ± 438.4 vs. 3224.6 ± 441.6 or 3241.7 ± 426.5, respectively; p = 0.013), suggesting the direct effect of maternal binge drinking before pregnancy on birth weight. Concomitantly, the prevalence of macrosomia was significantly higher in offspring groups from women with binge drinking than those with never and non-binge drinking (7.5% vs. 2.9% or 3.2%, respectively; p = 0.002) (Table 4 and Fig 1A). In addition, offspring from women with binge drinking had a significantly higher prevalence of admissions to neonatal intensive care unit than those with never or non-binge drinking (14.2% vs. 13.6% or 9.9%, respectively; p = 0.012). However, there were no difference in gender, height, head circumference, glucose, the prevalence of congenital anomaly, and apgar scores for 1 and 5 minutes between offspring from women with binge drinking and never or non-binge status. Meanwhile, to confirm the effect of pre-

**Table 3. Obstetric outcomes according to maternal alcohol-drinking status.**

| | All participants (n = 2,886) | Never drinking (n = 561) | Ever drinker[†] | | p-value |
|---|---|---|---|---|---|
| | | | Non-binge drinking (n = 2,099) | Binge drinking (n = 226) | |
| Gestational age (weeks) | 38.9 ± 1.5 | 38.8 ± 1.5 | 38.9 ± 1.5 | 39.1 ± 1.4 | 0.090 |
| Blood pressure at delivery | | | | | |
| Systolic blood pressure | 116.7 ± 11.2 | 117.2 ± 11.5 | 116.6 ± 11.1 | 116.6 ± 10.2 | 0.514 |
| Diastolic blood pressure | 72.8 ± 8.9 | 73.6 ± 9.3 | 72.6 ± 8.9 | 72.5 ± 8 | 0.060 |
| Preterm birth | | | | | |
| Yes (delivery at <37 weeks) | 146 (5.1) | 31 (5.5) | 106 (5.1) | 9 (4.0) | 0.670 |
| No (delivery at full term, ≥37 weeks) | 2740 (94.9) | 530 (94.5) | 1993 (95.0) | 217 (96.0) | |
| Gestational diabetes | | | | | |
| No | 2631 (93) | 503 (92.6) | 1933 (93.5) | 195 (89.5) | 0.074 |
| Yes | 197 (7.0) | 40 (7.4) | 134 (6.5) | 23 (10.6) | |
| Pregnancy-induced hypertension[‡] | | | | | |
| No | 2792 (98.7) | 536 (98.5) | 2040 (98.7) | 216 (99.1) | 0.883[‡] |
| Yes | 37 (1.3) | 8 (1.5) | 27 (1.3) | 2 (0.9) | |
| Perinatal depression[§] | | | | | |
| At 1st trimester | | | | | |
| No | 2274 (81.2) | 426 (81.5) | 1682 (81.9) | 166 (74.4) | **0.026** |
| Yes | 526 (18.8) | 97 (18.6) | 372 (18.1) | 57 (25.6) | |
| At 2nd trimester | | | | | |
| No | 2332 (86.8) | 433 (87.5) | 1724 (87.1) | 175 (82.9) | 0.215 |
| Yes | 354 (13.2) | 62 (12.5) | 256 (12.9) | 36 (17.1) | |
| At 3rd trimester | | | | | |
| No | 2150 (86.8) | 417 (90.1) | 1579 (86.4) | 154 (83.2) | **0.036** |
| Yes | 326 (13.2) | 46 (9.9) | 249 (13.6) | 31 (16.8) | |
| Postpartum depression[§] | | | | | |
| No | 1750 (84.5) | 322 (85.4) | 1309 (85.5) | 119 (73.5) | **0.000** |
| Yes | 320 (15.5) | 55 (14.6) | 222 (14.5) | 43 (26.5) | |
| Complication during delivery[¶] | | | | | |
| No | 2566 (88.9) | 509 (90.7) | 1861 (88.7) | 196 (86.7) | 0.211 |
| Yes | 320 (11.1) | 52 (9.3) | 238 (11.3) | 30 (13.3) | |
| Delivery type | | | | | |
| Vaginal delivery | 1765 (61.2) | 335 (59.7) | 1299 (61.9) | 131 (58.0) | 0.381 |
| Cesarean delivery | 1121 (38.8) | 226 (40.3) | 800 (38.1) | 95 (42) | |

Data are expressed as mean ± SD or n (%). The p-value is a comparison between the three groups. Bold values are statistically significant findings (p<0.05).

[†]Ever drinker included former (n = 2,322) and current drinker (n = 3). [‡]p value is calculated by Fisher's exact test.

[‡]Pregnancy-induced hypertension was defined by a systolic blood pressure ≥140 mmHg and/or diastolic blood pressure ≥90 mmHg without proteinuria (<0.3 g in a 24-hour urine collection) and the hypertension must have developmed after 20 weeks of gestation.

[§]Perinatal/postpartum depression were defined by a score of ≥10 on K-EPDS (Modified Korean-Edinburgh Postnatal Depression Scale) during pregnancy or in the 4 weeks following delivery, respectively.

[¶]Complication including shoulder dystocia, injuries of parturient canal, abruption placentae, premature rupture of membranes, uterine rupture and eclampsia.

pregnancy drinking on the offspring's characteristics and outcomes, 2,554 participants excluding 332 women who drank alcohol in the first trimester were analyzed. As shown in S1 Table, compared with the results analyzed in 2,886 participants (Table 4), there was little difference in the prevalence of macrosomia and other outcomes. To further confirm the effects of maternal

**Table 4. Offspring's characteristics and outcomes according to maternal alcohol-drinking status before pregnancy.**

| | All participants (n = 2,886) | Never drinking (n = 561) | Ever drinker[†] | | p-value |
|---|---|---|---|---|---|
| | | | Non-binge drinking (n = 2,099) | Binge drinking (n = 226) | |
| Gender | | | | | |
| Male | 1479 (51.2) | 293 (52.2) | 1064 (50.7) | 122 (54.0) | 0.562 |
| Female | 1407 (48.8) | 268 (47.8) | 1035 (49.3) | 104 (46.0) | |
| Weight (g) | 3244.7 ± 430.9 | 3224.6 ± 441.6[a] | 3241.7 ± 426.5[a] | 3322.7 ± 438.4[b] | **0.013** |
| Height (cm) | 49.6 ± 2.3 | 49.5 ± 2.2 | 49.6 ± 2.1 | 49.7 ± 4 | 0.431 |
| Head circumference (cm) | 34.5 ± 1.5 | 34.4 ± 1.4 | 34.5 ± 1.3 | 34.5 ± 2.7 | 0.228 |
| Glucose (mg/dl)[‡] | 82.9 ± 18.9 | 80.5 ± 16.9 | 83.6 ± 19.5 | 81.5 ± 17.2 | 0.099 |
| Macrosomia | | | | | |
| No | 2786 (96.5) | 545 (97.1) | 2032 (96.8) | 209 (92.5) | **0.002** |
| Yes | 100 (3.5) | 16 (2.9) | 67 (3.2) | 17 (7.5) | |
| Congenital anomaly | | | | | |
| No | 2835 (98.2) | 550 (98.0) | 2061 (98.2) | 224 (99.1) | 0.636[§] |
| Yes | 51 (1.8) | 11 (2.0) | 38 (1.8) | 2 (0.9) | |
| Admissions to neonatal intensive care unit | | | | | |
| No | 2571 (89.1) | 485 (86.5) | 1892 (90.1) | 194 (85.8) | **0.012** |
| Yes | 315 (10.9) | 76 (13.5) | 207 (9.9) | 32 (14.2) | |
| Apgar score | | | | | |
| 1 minute, mean | 7.97 ± 0.8 | 7.93 ± 0.7 | 7.97 ± 0.7 | 7.91 ± 0.8 | 0.359 |
| 5 minute, mean | 8.80 ± 0.7 | 8.76 ± 0.6 | 8.80 ± 0.6 | 8.72 ± 0.7 | 0.233 |

Data are expressed as mean ± standard deviation (SD) or n (%). The p-value is a comparison between the three groups. Bold values are statistically significant findings (p<0.05).

[a,b]Different letters represent statistical difference by Tukey's multiple comparison test.

[†]Ever drinker included former (n = 2,322) and current drinker (n = 3).

[‡]Only 1,039 offspring were included in the analysis.

[§]The p-value is calculated by Fisher's exact test.

alcohol-drinking before pregnancy on birth weight or macrosomia development in offspring, the participants were re-categorized into 5 groups based on the number of drinks (cup or ounce) consumed per month (never drinking, ≤10 cup (4.2 oz), 10 cup (4.2 oz)< - ≤20 cup (8.5 oz), 20 cup (8.5 oz)< - ≤ 30 cup (12.7 oz), and >30 cup (12.7 oz). The absolute frequency of macrosomia was highest in women with >30 cup (12.7 oz) drinking (5.5%) and lowest in those with never drinking (2.9%) (Cochran-Armitage trend test; $p = 0.031$) (Fig 1B). However, the mean birth weight for each offspring group was not significantly increased depending on the number of drinks (cup or oz) per month and their correlations were not statistically significant (Pearson's correlation analysis; $r = 0.048$, $p = 0.281$).

## Independent association between maternal alcohol drinking before pregnancy and macrosomia development

When assessing the relative risk of significant macrosomia predicted by maternal alcohol-drinking status (Table 5), we found that the unadjusted odds ratio (OR) for developing macrosomia in women with binge drinking was significantly increased compared with those with never drinking as reference groups (OR = 2.77; 95% CI 1.37 to 5.59, $p = 0.004$). Next, to adjust for confounding covariates that affect the prevalence of macrosomia, we applied three multivariable logistic regression models. Women with binge drinking before pregnancy had a higher risk for developing macrosomia in a minimally adjusted model (model 1) using

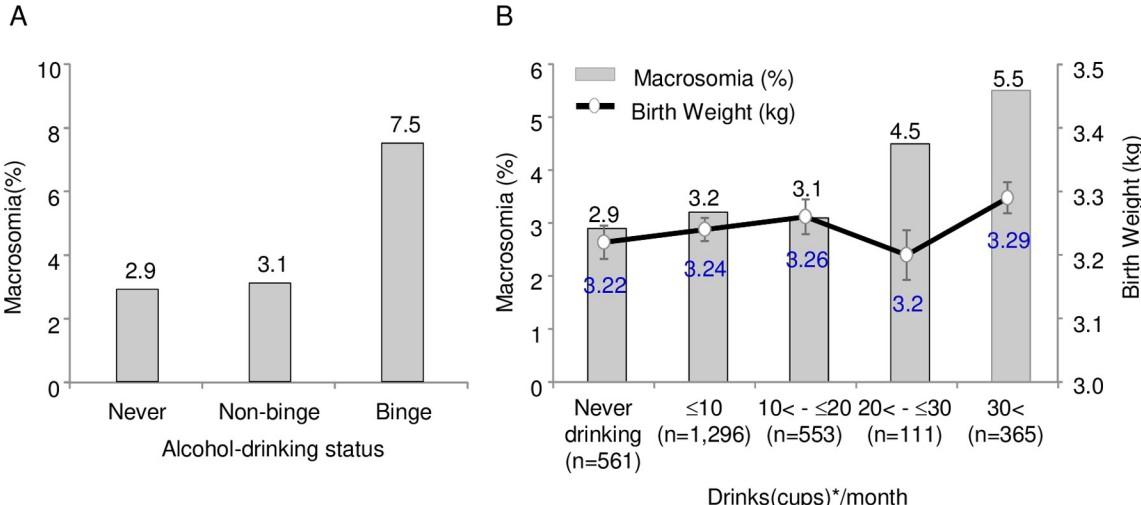

**Fig 1. The prevalence of macrosomia and birth weight according to maternal alcohol drinking before pregnancy.** (A) Comparison of the prevalence (%) of macrosomia in participants (n = 2,886) with different alcohol-drinking status (*p* value was determined by the chi-square test). (B) Difference in the prevalence of macrosomia (bar graph) and birth weight (linear graph) according to the number of drinks (cups) consumed per month. Changes of macrosomia (*p* = 0.031) and birth weight ($\gamma$ = 0.048, *p* = 0.281) in offspring groups classified by the number of drinks (cups) was determined via Cochran-Armitage trend test and Pearson's correlation analysis, respectively. *Drinks (cups) may be converted to the unit of volumes or weight (ounce, oz): 10 drinks(cups), 4.2 oz; 20 drinks (cups), 8.5 oz; 30 drinks (cups), 12.7 oz.

maternal age, education, and marital status (adjusted OR = 2.88; 95% CI 1.42 to 5.84, *p* = 0.003) compared with those with never drinking. When we further adjusted for other variables (model 1 variables plus smoking and physical activity for model 2), the independent association was consistently maintained (adjusted OR = 2.85; 95% CI 1.38 to 5.89, *p* = 0.005). As expected, when other well-established risk factors of macrosomia (model 2 variables plus gestational age, pre-pregnancy body mass index, parity, offspring's gender and gestational diabetes for model 3) were taken into account, women with binge drinking remained statistically and clinically significant (adjusted OR = 2.29; 95% CI 1.08 to 4.86, *p* = 0.031). Also, similar

**Table 5. Odds ratio with 95% CIs of macrosomia depending on maternal alcohol-drinking status before pregnancy.**

| | No. of subjects | Never drinking | Ever drinker† | | | |
|---|---|---|---|---|---|---|
| | | | Non-binge drinking | | Binge drinking | |
| | | | OR (95% CI) | *p* value | OR (95% CI) | *p* value |
| Macrosomia (>4,000g) | | | | | | |
| Unadjusted | 2,886 | 1.00 | 1.12 (0.65–1.95) | 0.681 | 2.77 (1.37–5.59) | 0.004 |
| Model 1 | 2,886 | 1.00 | 1.13 (0.65–1.97) | 0.669 | 2.88 (1.42–5.84) | 0.003 |
| Model 2 | 2,801 | 1.00 | 1.08 (0.62–1.89) | 0.787 | 2.85 (1.38–5.89) | 0.005 |
| Model 3 | 2,746 | 1.00 | 1.01 (0.57–1.80) | 0.968 | 2.29 (1.08–4.86) | 0.031 |

We assessed the ORs depending on alcohol-drinking status for offspring macrosomia using multivariable logistic regression analyses. Data are OR (95% CI) for unadjusted and adjusted models 1–3.

Model 1 adjusted for maternal age, education and monthly income

Model 2 adjusted for maternal age, education, monthly income, smoking and physical activity

Model 3 adjusted for maternal age, education, monthly income, smoking and physical activity, gestational age, pre-pregnancy body mass index, parity, offspring's gender and gestational diabetes

† Ever drinker included former (n = 2,322) and current drinker (n = 3). OR, odds ratios; CI, confidence interval.

results were obtained in analysis for 2,554 participants excluding 332 women who drank alcohol during the first trimester of pregnancy (S2 Table) and 2,746 participants who had valid data for all potential confounders (S3 Table), respectively.

We further assessed the multivariable-adjusted odds ratio of developing macrosomia for each risk factor (S4 Table). In multivariable-adjusting analyses for each risk factor, maternal binge drinking has the greatest risk of developing macrosomia and followed by offspring gender-boys (adjusted OR = 2.04; 95% CI 1.32 to 3.17, $p$ = 0.001), gestational age (adjusted OR = 1.92; 95% CI 1.57 to 2.35, p<0.0001), gestational diabetes (adjusted OR = 1.90; 95% CI 1.02 to 3.55, p = 0.043), and pre-pregnancy BMI (adjusted OR = 1.14; 95% CI 1.07 to1.21, p<0.0001). These results suggest that maternal binge drinking before pregnancy is associated with offspring's macrosomia independently of traditional risk factors for developing macrosomia.

## Maternal binge drinking before pregnancy may be an independent index to predict the risk of macrosomia

In a sub-analysis, to investigate whether maternal binge drinking may also affect stratified risk factor-mediated macrosomia, we classified each risk factor into low- and high-risk groups according to the severity of each risk. Then, stratified analysis using maternal alcohol-drinking status before pregnancy provided additional discrimination for the risk of macrosomia (S2 Fig). Among women with high-risk of each risk factor, the prevalence of macrosomia for binge drinking were significantly increased compared to those for never or non-binge drinking. Interestingly, if women has binge drinking status, the prevalence of macrosomia was also significantly increased even in all groups with low-risk such as maternal age <35, BMI <25, nulliparous, former or current smoking, more than moderate physical activity, and no GDM; although their prevalence levels were more less than those of women with high risk. Indeed, differ to incremental changes in discrimination for binge drinking in high-risk groups for most of risk factors, these discriminable changes for parity and GDM risks were not significant compared to those of low-risk groups. As well, although women with high-risk of maternal age, prepregnancy BMI, and GDM have just non-binge drinking status, the prevalence for macrosomia tend to increase compared to those of low-risk women. Next, to assess the discrimination ability of maternal binge drinking before pregnancy in predicting the development of neonatal macrosomia, we obtained the area under the receiver operating characteristic (AUROCs) curves for the conventional model with all traditional risk factors for macrosomia and our new model including binge drinking before pregnancy with conventional model. The AUROC were 0.778 (95% CI, 0.737 to 0.819) for the conventional model and 0.784 (95% CI, 0.743 to 0.825) for new model with binge drinking (S3 Fig). The improvements of AUROC of new model compared to the conventional model (Δ = 0.006; 95% CI, -0.007 to 0.018; P = 0.4062) were not significant. To evaluate improvements in the reclassification by addition of the binge drinking before pregnancy to the conventional model, the net reclassification improvement (NRI) were calculated (Table 6).

The results show that our new model adding the binge drinking to the conventional models led to significant improvements of 10.6% (95% CI, 2.03 to 19.07; 8.3% for cases plus 2.3% for non-cases) in NRI, which examines correct and incorrect movements between the user-specified risk categories. Taken together, these results indicating that binge drinking before pregnancy may be an independent biomarker to predict the risk of macrosomia both in women with low-risk and high-risk status.

## Discussion

As a result of analysis using the Korean pregnancy-registry database, it was found that there was a significant relationship between maternal binge drinking before pregnancy and the

**Table 6. Reclassification of predicted risk among participants who developed macrosomia and those who do not developed macrosomia after follow-up.**

| Estimated risk(conventional model)[a] | Estimated risk (new model)[a] | | | | Reclassified[b] | | Net correctly reclassified (%)[c] |
|---|---|---|---|---|---|---|---|
| | Low (<2%) | Mid-low (2% to 4%) | Mid-high (4% to 11%) | High (>11%) | Increased | Decreased | |
| Macrosomia (n = 97) | | | | | | | |
| Low (<2%) | 6 | 1 | 0 | 0 | 13 | 5 | 8.3 |
| Mid-low (2% to 4%) | 2 | 19 | 4 | 0 | | | |
| Mid-high (4% to 11%) | 0 | 3 | 38 | 8 | | | |
| High (>11%) | 0 | 0 | 0 | 16 | | | |
| Non-Macrosomia (n = 2649) | | | | | | | |
| Low (<2%) | 1133 | 44 | 3 | 0 | 127 | 188 | 2.3 |
| Mid-low (2% to 4%) | 97 | 603 | 53 | 0 | | | |
| Mid-high (4% to 11%) | 0 | 73 | 509 | 27 | | | |
| High (>11%) | 0 | 0 | 18 | 89 | | | |
| NRI (95% CI) | | | | | | | 10.6 (2.03 to 19.07) |

[a]The estimated risk of the two models (conventional and new model) were categorized into 4 groups with different cutoffs. The cutoffs were classified by the definitions of low, mid-low, mid-high, and high based on the deciles of the distribution of absolute risk for macrosomia and the NRI statistics in various numbers of the intervals (2~5%) and various cut-off points of high risk (from 10 to 15% by 1%) were tested. Conventional model includes gestational age (weeks), pre-pregnancy body mass index (BMI) (kg/m$^2$), parity (the number of deliveries), newborn's gender and gestational diabetes (yes/no); new model includes binge drinking before pregnancy plus conventional model.

[b] & [c]Reclassification improvement is 8.3% for cases ([13–5]/97), while reclassification improved in non-cases by 2.3% ([188–127]/2649), leading a net-reclassification-improvement of 10.6%. NRI = net reclassification improvement; CI, confidence interval.

development of macrosomia in offspring. The data also showed that binge drinking before pregnancy can have a crucial effect on the development of macrosomia independent of traditional risk factors and may be an independent indicator to predict the risk of macrosomia both in women with low- and high-risk status. A lot of evidence about the harmful effects of drinking during pregnancy on maternal and prenatal health have been continuously accumulating, whereas the effects and impacts of pre-pregnancy drinking on the progressive development of the fetus and postnatal growth remain obscure. Additionally, whether there are negative effects of drinking on women, especially women of childbearing age, remains unclear and there is little research about the relationship between alcohol drinking before pregnancy and postnatal macrosomia. Here, in our study using the Korean pregnancy registry, we demonstrated that subjects with binge drinking status, but not in those with low-moderate drinking, before pregnancy had an approximately 2.29-fold increased risk of significant macrosomia, independently of traditional risk factors for macrosomia such as maternal age, prepregnancy BMI, parity, gestational age, and gestational diabetes. As well, the prevalence of macrosomia in women with binge drinking before conception was specifically potentiated in women with high-risk such as prepregnancy obesity, non-smoker, low exercise in prepregnancy, maternal age ≥35, and multiparity, suggesting that maternal binge drinking before pregnancy can make women more susceptible to those exposed to these risk factors, and thus vulnerable to the incidence of macrosomia. Although an in-depth mechanism for the independent association between the status of maternal alcohol drinking and offspring macrosomia by using the clinical samples (serum, tissue, or urine) was not elucidated in this study, our results firstly provided evidence with positive significant association between maternal binge drinking habit before pregnancy and offspring macrosomia.

In the multiple previous literatures, various risk factors of macrosomia or high birth weight have been suggested [24–31]. Maternal age, higher parity, pre-pregnancy obesity, gestational diabetes, history of previous macrosomic infant delivery, post-term pregnancy and infant gender (male) are all positively associated with macrosomia. Although several researchers have suggested an effect of maternal alcohol consumption during pregnancy on newborn's birth weight, evidence for an association between alcohol intake in pre-pregnancy and macrosomia remains scarce. In addition, the effects of alcohol drinking during pregnancy on offspring's birth weight are still controversial. Some studies demonstrated that alcohol consumption during pregnancy is independently associated with an increase in low birth weight [46, 47], whereas other studies suggested that there was no impact of newborns small for gestational age or preterm birth [48]. In a recent non-human primate study of alcohol consumption, there was no significant difference in fetal birthweight at time of delivery in ethanol-exposed fetus compared with control animals [49]. The inconsistency of these results could be due to differences in race/ethnicity, study design, definitions of exposure and outcome, and environmental factors for each study. Although not a clinical data-based research, we recently reported that in mice exposed to ethanol for 2-weeks before pregnancy, postnatal birth weight was approximately two-fold higher in pups of ethanol-fed mice than in those of pair-fed mice, which correlated with postnatal growth retardation [17]. This macrosomia phenomenon differs from previous reports that ethanol-exposed infants have lower birth weights than those of control group [50]. This discrepancy may be due to the time and duration of exposure to ethanol, such as before or during pregnancy. On the other hand, several previous studies demonstrated that offspring birth weight is associated with second- and third-trimester postprandial blood glucose levels, but not with fasting or mean glucose levels [51], suggesting that maternal homeostasis on glucose and insulin tolerance in mid- or late-pregnancy period may be required for the normal development of the fetus and infant. This possibility was strongly supported by our previous mouse study showing that alcohol drinking before, not during, pregnancy was closely associated with the alteration on maternal homeostasis on glucose and insulin tolerance during the progression of pregnancy [17]. To the best of our knowledge, this is the first evidence to provide adverse effects of alcohol drinking before pregnancy on postnatal macrosomia and offspring's growth retardation in an *in vivo* mouse model. Furthermore, these adverse effects of pre-pregnancy drinking on birth weight were apparently reinforced by our current study using clinical data based on Korean Pregnancy Registry Cohort. Lab-clinical data also show that women with binge drinking pattern before pregnancy exhibited significant increases of third-trimester fasting glucose, not in first-trimester, compared to non-binge drinking groups. The elevation of fasting glucose in binge drinking groups in the third trimester was correlated with significant increases in hemoglobin, hematocrit, ALT, total protein, and albumin levels compared to those of non-binge drinking groups. Our data provides solid evidence for an independent association between macrosomia and alcohol drinking before pregnancy regardless of the influence of traditional risk factors that can affect the development of macrosomia. Furthermore, our further analysis for 2,554 participants excluding 332 women who drank alcohol during the first trimester (S2 Table) and 2,746 participants who had valid data for all potential confounders (S3 Table) clearly confirmed the role of pre-pregnancy drinking as an independent risk factor for the development of macrosomia.

In fact, it is widely accepted that early stage of pregnancy is considered as an important period to prepare maternal metabolic homeostasis and energy metabolism for demands of the fetal development or growth [16]. So, changes in maternal food intake and physical activity behavior before pregnancy or during early pregnancy may alter energy and nutrient metabolism available for fetal growth, making them vulnerable to various stressors such as obesity, smoking, alcohol intake and drug intake. In particular, acute or chronic alcohol consumption

before pregnancy may affect the first adjustment of maternal nutrient or energy metabolism and may thus trigger oxidative stress-mediated metabolic disorders. All major organs begin to form and develop in the early stages, which is called the prenatal development period, and thereafter, during the perinatal period, fetal development and maturation are continued [17, 52], suggesting that the fetal body and organs are developing throughout pregnancy and can be affected by exposure to alcohol at any time. In particular, since the limbs, eyes, and ears are being formed at the fourth week of gestation in humans, the effects of alcohol consumption in early pregnancy can cause defects in these systems and organs [17]. Consistently, our previous study suggested that the mice exposed to alcohol before pregnancy displayed the retardation on eye development that correlated with impaired glucose and insulin metabolism [17]. In addition, our previous reports demonstrated that ethanol-fed mice were closely associated with the alteration of glucose and insulin metabolism, which is strongly related to the development of type 2 diabetes through pancreatic β-cell dysfunction and apoptosis [53, 54]. Based on these results, we can propose the possibility that maternal alcohol drinking before or during early pregnancy may be involved in the altered regulation of maternal metabolic homeostasis, leading to impaired fetal development and child's growth retardation. On the other hand, a recent study demonstrated that binge eating before or during pregnancy is associated with prematurity, macrosomia, and future risk of diabetes and metabolic syndrome in infants as well as with higher gestational weight gain and greater postpartum weight retention in mother [55]. In addition, some studies demonstrated that it is not only the type of diet (i.e., frequency of fat intake) but also the type of eating behavior (i.e., binge eating) that seems to contribute to explaining binge drinking [56]. Along the same line, binge eating behaviors may be associated with binge drinking and could be a gateway to the initiation and escalation of binge drinking, resulting in an increased risk of macrosomia. Nevertheless, studies on pregnancy and neonatal outcomes among women with ongoing or previous eating disorders are scare. Unfortunately, our current study did not take into account the relationship between binge eating (or eating disorders) and binge drinking (and its-mediated macrosomia). However, it may be important to investigate the relationship between eating patterns before pregnancy and binge drinking behavior, which may result in an increased risk of macroaomia. Therefore, it may be necessary to establish additional data by requesting the additive survey examinations and evaluations on eating disorders (binge eating) in the Korean pregnancy registry cohort for future study. On the other hand, although previous several studies demonstrated that maternal smoking was associated with decreased risk of macrosomia, the association of maternal smoking with infant weight loss and even reduced macrosomia remains unclear. Most of early studies reported that maternal smoking was associated with decreased risk of macrosomia [40–42], but recent studies found no crude or adjusted association between maternal smoking and macrosomia [57–59]. Our data clearly exhibited that there was no difference in the risk of developing macrosomia between the high-risk non-smoking group and the low-risk group for former or current smoking (S4 Table). However, when both groups for former or current smoking and nonsmoking were exposed to binge drinking before pregnancy, the risk of macrosomia was significantly increased into 6.8 and 7.9 folds, respectively (S2A Fig), suggesting the vulnerable effect of binge drinking before pregnancy.

There is currently no cure for macrosomia, and it is difficult to estimate or predict a baby's birth weight in advance. A definitive diagnosis and prognosis for fetal and postnatal macrosomia, respectively, cannot be made until after the baby is born and weighed. Because the prognosis of macrosomia always ends with serious long-term clinical outcomes such as metabolic complications and growth retardation in whole life-span [60], a practical and effective solution to the occurrence of these complications is its prevention. Therefore, it is absolutely necessary to identify new risk factors that can improve the accuracy of early prediction and diagnosis of

macrosomia, and develop a novel risk prediction model that applies them. Our results suggest that binge drinking before pregnancy is an independent risk factor for the prediction of incident macrosomia. As a result of confirming the predictive power of the new risk models including maternal binge drinking before pregnancy through AUROCs, it was similar to that of conventional risk prediction model with all traditional risk factors. However, when applying our risk model to other definitions of macrosomia using NRI analysis that user-specific categorized the estimated risk into 4 levels, the reclassification ability was significantly improved by 10.6% (95% CI, 2.03 to 19.07; p = 0.0006). Moreover, in a multivariable logistic regression model, maternal binge drinking before pregnancy was associated with a significantly higher risk of macrosomia compared to traditional modifiable risk factors such as prepregnancy BMI and gestational diabetes.

Our study has some limitations. First, although multiple plausible factors have been considered and controlled, we cannot be fully ruled out the possibility that our findings may have been affected by unmeasured or unknown residual confounding. Nevertheless, to investigate the independent effects of maternal drinking before pregnancy on the development of macrosomia, we built diverse and step-by-step models, and adjusted for previously well-known major risk factors for macrosomia, including gestational diabetes and lifestyle variables. Second, it is not possible to calculate the exact amount of alcohol intake because the type of alcoholic beverage (eg, beer, soju, wine, spirits, etc) are not examined. Therefore, whether there is a dose-response relationship between quantity of alcohol and macrosomia was not determined. However, binge drinking was used as an exposure variable for the assessment of alcohol intake, and binge drinking has been generally used in epidemiological studies as a definition without considering the type of alcohol. In additions, the definition of binge drinking included not only the amount (cup) but also the frequency of drinking. In some cases, this can be more useful information than an absolute quantity variable. Third, our cohort's information on maternal alcohol consumption were collected via maternal self-report according to each specific questionnaire or interview-based questionnaires, which could have missing data and led to potential bias. Particularly with respect to smoking and alcohol consumption before or during pregnancy, self-reports of substance use may have underestimated actual use due to the negative perception and stigmatization. In fact, the questionnaire on alcohol drinking included lifetime drinking and past 1 year and 6 months, current drinking, duration, and amount; however, data on the frequency and amount of drinking for the past 1 year and 6 months were not accurately collected due to related data missing. So, our study used survey and analysis data for lifetime drinking instead of those for past 1 year and 6 months. Meanwhile, another pregnancy cohort that is currently being constructed has more accurate data than the existing cohort, future studies using these data will be able to provide more accurate and specific results than now. Despite these limitations, this is the first study to investigate the association between macrosomia and pre-pregnancy drinking status. In fact, there is currently no worldwide consensus on how many drinks constitute "binge drinking", but in the United States, academic studies have defined the term to mean consuming five or more standard drinks (male), or four or more drinks (female), over a two-hour period [61]. Alcohol consumption varies widely across countries, population groups and time periods, depending on the political and social environment [62, 63]. In addition, the definition of binge drinking and the size of a standard drink vary widely between and even within countries. As well, since all subjects participated in this study are Korean, no interracial comparison results were presented.

Despite the limitations discussed above, the strengths of this study is to providing the direct evidence that maternal drinking before pregnancy, but not during pregnancy, is closely associated with the development of macrosomia in offspring using a Korean pregnancy registry

database (n = 2,886). Our current study also confirmed our previous results showing the adverse impact of maternal drinking before pregnancy on impaired fetal development and postnatal macrosomia by using animal models [17]. Our analytical results also provided clear evidence that maternal binge drinking before pregnancy correlates with an increased risk for incident macrosomia and may serve as an independent risk factor predicting the incident risk of macrosomia in women. As well, compared to previously suggested risk factors for macrosomia, our new model achieves similar (in AUROC curves) or improved (NRI category) predictive power, uses readily available preprocedural factors, and is timely preprocedural risk prediction generally has many potential benefits. These results could help public health or clinical intervention working groups to establish national or individual tailored procedures, such as specific preventive strategies, as well as health policies or campaigns regarding the risk or life-style modification for alcohol drinking before pregnancy. Moreover, our previous studies using mice fed ethanol before pregnancy supported a deleterious effect of maternal alcohol consumption before pregnancy on fetal development. Although we provide solid evidence for an independent association between macrosomia and maternal alcohol drinking before pregnancy regardless of the influence of traditional risk factors that can affect the development of macrosomia, in-depth mechanisms and target molecules for the independent association between maternal drinking status and macrosomia using clinical samples such as serum, tissue, or urine, were not shown here. However, we can clearly propose that maternal binge drinking before pregnancy is an adverse threat for the development of infant's macrosomia, which is closely associated with the adverse outcomes of infant's future health, such as obesity, chronic disease, etc. Therefore, to prevent this prevalent binge drinking of women and to minimize the associated risks is the most effective strategy for reducing the transfer of adverse outcomes from pregnant mothers into the infants and child.

Taken together, we provided evidence that binge drinking before pregnancy was associated with a significantly higher risk for offspring's macrosomia and it may be an independent risk factor to predict the risk of macrosomia regardless of the presence or absence of traditional risk factors for macrosomia. Finally, to ensure the health of the mother and the fetus during pregnancy, it is proposed to establish a public health policy for the reduction or prevention of drinking before pregnancy.

## Supporting information

**S1 Fig. Flow diagram of subject inclusion and exclusion in the Korean pregnancy registry cohort.** Of the total subjects (n = 4,542), 2,886 who had complete follow-up data were finally included.
(TIF)

**S2 Fig. Maternal alcohol drinking before pregnancy is closely associated with macrosomia.** (A-F) Prevalence of macrosomia according to the maternal alcohol drinking status before pregnancy and the presence or absence of traditional risk factors for macrosomia. Stratified analysis using the severity of maternal alcohol drinking status allowed for the prediction of risk of developing macrosomia in women with or without high-risk of traditional risk factors for macrosomia. Traditional risk factors were classified according to following as: Smoking (Former or current, low-risk; None, high-risk), Physical activity (More than moderate, low-risk; None or light, high-risk), Maternal age (<35, low-risk; ≥35, high-risk), Prepregnancy BMI (<25, low-risk; ≥25, high-risk), Parity (nulliparous, low-risk; ≥1, high-risk), GDM (No, low-risk; Yes, high-risk). §There is no subjects with GDM, who has binge drinking status (n = 0).
(TIF)

**S3 Fig. Comparison of the area under the receiver operating characteristic curves (AUROCs) between two prediction models with or without a binge drinking in predicting developed macrosomia.** The AUROCs of two models (new and traditional) were 0.784 (95% CI, 0.743 to 0.825) and 0.778 (95% CI, 0.737 to 0.819), respectively. The estimate for difference of two AUROCs was 0.005 (95% CI, -0.007 to 0.0181; p = 0.4062).
(TIF)

**S1 Table. Offspring's characteristics and outcomes according to maternal alcohol-drinking status before pregnancy in 2,554 participants excluding 332 women who drank alcohol in the first trimester (related to Table 4).**
(DOCX)

**S2 Table. Odds ratio with 95% CIs of macrosomia depending on maternal alcohol-drinking status before pregnancy in 2,554 participants excluding 332 women who drank alcohol in the first trimester (related to Table 5).**
(DOCX)

**S3 Table. Odds ratio with 95% CIs of macrosomia depending on maternal alcohol-drinking status before pregnancy in 2,746 participants who had valid data for all potential confounders (related to Table 5).**
(DOCX)

**S4 Table. Multivariable-adjusted ORs of developing macrosomia for the risk factors of macrosomia.**
(DOCX)

## Acknowledgments

We thank all the staff of the Division of Material fetal Medicine in Cheil General Hospital and CHA hospital who were involved in the conduction of the Korean pregnancy registry cohort. And also, the authors express gratitude to the pregnant women who participated in the Korean pregnancy registry cohort study and the members of KNIH's Woman Health Research Team who qualified and curated the data and made it publicly available.

## Author Contributions

**Conceptualization:** Seul Koo, Hyun Young Park, Won-Ho Kim.

**Data curation:** Seul Koo, Ji Yeon Kim, Ji Hye Park, Nam Kyoo Lim, Won-Ho Kim.

**Formal analysis:** Seul Koo, Ji Yeon Kim, Ji Hye Park, Nam Kyoo Lim, Won-Ho Kim.

**Funding acquisition:** Hyun Young Park, Won-Ho Kim.

**Investigation:** Seul Koo, Ji Yeon Kim, Won-Ho Kim.

**Methodology:** Seul Koo, Gu Seob Roh, Won-Ho Kim.

**Project administration:** Seul Koo, Ji Yeon Kim, Hyun Young Park, Won-Ho Kim.

**Resources:** Seul Koo, Gu Seob Roh, Won-Ho Kim.

**Software:** Seul Koo, Nam Kyoo Lim.

**Supervision:** Hyun Young Park, Won-Ho Kim.

**Validation:** Seul Koo, Ji Yeon Kim, Ji Hye Park, Nam Kyoo Lim, Won-Ho Kim.

**Visualization:** Seul Koo, Ji Yeon Kim, Nam Kyoo Lim, Won-Ho Kim.

**Writing – original draft:** Seul Koo, Nam Kyoo Lim, Won-Ho Kim.

**Writing – review & editing:** Seul Koo, Ji Yeon Kim, Ji Hye Park, Gu Seob Roh, Hyun Young Park, Won-Ho Kim.

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
