## [Decision Letter · Decision Letter 0]

13 Jan 2022

PONE-D-21-31508Binge alcohol drinking before pregnancy is closely associated with the development of macrosomia: Korean pregnancy registry cohortPLOS ONE

Dear Dr. (KNIH),

Thank you for submitting your manuscript to PLOS ONE. After careful consideration, we feel that it has merit but does not fully meet PLOS ONE’s publication criteria as it currently stands. Therefore, we invite you to submit a revised version of the manuscript that addresses the points raised during the review process.

While the reviewers overall feel the paper has merit, they also provided a large number of specific suggestions that the authors should address.  In addition, we recommend additional English-language editing.==============================

We look forward to receiving your revised manuscript.

Kind regards,

Emily W. Harville

Academic Editor

PLOS ONE

Journal Requirements:

2. Please provide additional details regarding participant consent. In the ethics statement in the Methods and online submission information, please ensure that you have specified whether consent was written or verbal/oral. If consent was verbal/oral, please specify: 1) whether the ethics committee approved the verbal/oral consent procedure, 2) why written consent could not be obtained, and 3) how verbal/oral consent was recorded. If your study included minors, please state whether you obtained consent from parents or guardians in these cases. If the need for consent was waived by the ethics committee, please include this information.

Reviewers' comments:

Reviewer's Responses to Questions

**Comments to the Author**

1. Is the manuscript technically sound, and do the data support the conclusions?

Reviewer #1: Yes

Reviewer #2: Yes

2. Has the statistical analysis been performed appropriately and rigorously? 

Reviewer #1: Yes

Reviewer #2: I Don't Know

3. Have the authors made all data underlying the findings in their manuscript fully available?

Reviewer #1: Yes

Reviewer #2: Yes

4. Is the manuscript presented in an intelligible fashion and written in standard English?

Reviewer #1: Yes

Reviewer #2: Yes

5. Review Comments to the Author

Reviewer #1: The manuscript “Binge alcohol drinking before pregnancy is closely associated with the development of macrosomia: Korean pregnancy registry cohort” examined the effects of binge alcohol drinking on birth weight. They analyzed 4542 births in the Korean pregnancy registry enrolled between 2013 and 2017 and found that binge drinking before pregnancy increased the risk of developing macrosomia. The authors did a great job analyzing the associations between alcohol intake (measured in different ways) and macrosomia and presenting their results step by step. More importantly, the authors adjusted for risk factors associated with both macrosomia and live birth, which can help address the live birth bias issue. My biggest concern is related to residual confounding (such as binge eating/eating disorders). Nevertheless, the study provides valuable evidence to study the effect of pre-conception behaviors on perinatal outcomes.

Method:

1. I recommend giving more information about the study population. For example, how and where (hospital?) were pregnant women recruited into the registry? The registry cohort seems to target all pregnant women in Korea, but it only recruited 4542 pregnant women between 2013-2017. This is important because it determines whether the study sample is a representative sample of the general population.

2. “Of those, we initially selected 3,472 pregnancies with singleton and complete follow-up data (exclude those with follow-up loss (n=1,021) and multiple pregnancy (n=49))”. Were the 3,472 pregnancies those who had valid (non-missing) data on both binge drink and macrosomia? The inclusion criteria in this step should be provided in the text.

3. I recommend providing the reasons for excluding participants with pre-existing disease before pregnancy.

4. In the following sentence:

“If women answered ‘Former drinker’ or ‘Current drinker’, they were then further asked average frequency of alcohol-drinking (≤1/month, 1-2/month, 2-3/week, ≥4/week, everyday) and quantity of alcohol-drinking per drinking day (1-2 drinks, 3-4 drinks, 5-6 drinks, 7-9 drinks, ≥10 drinks).”

Were pregnant women asked about their average frequency of alcohol consumption in their lifetime? in the past year? 6 months? Or 1 month?

5. In the “definition of macrosomia and its risk factors” section, I recommend specifying whether “smoking” and “physical activity” refer to smoking and physical activity before pregnancy or during pregnancy. Provide definitions of different levels of physical activity.

Results:

1. Include the association results between binge drinking and macrosomia, excluding 332 women who drank during the first trimester in the supplementary file.

2. Table 2 shows the lab results from 1st and 3rd trimesters.

a. Recommend specifying whether glucose results reflect fasting or non-fasting glucose level.

b. recommend specifying whether the p-values reflect the comparison between the never drinking and ever drinker group or the comparison across the three groups.

3. In table 3, the authors provided the results of the association between alcohol intake and obstetric outcome. Definitions of obstetric outcomes, such as pregnancy-induced hypertension and perinatal/postpartum depression, should be provided.

4. In table 5, the same group of participants should be used when comparing different models to make the results comparable. For example, I suggest the authors provide results of the unadjusted model for all 2886 participants and 2746 participants who had valid data for all potential confounders, respectively. Provide results of models 1-3 for the 2746 participants.

5. Table 6 compared the predictive performance of the conventional and new models.

a. List the factors included in both models in the text of the footnote of the table.

b. Describe how the cutoffs (low, mid-low, mid-high, high) were chosen.

Discussion:

1. In the second paragraph of this section, I recommend discussing the lab results in table 2 when discussing the association between alcohol drinking before pregnancy and maternal hemostasis.

2. I recommend discussing whether the authors evaluated binge eating /eating disorders among those participants. Binge eating/eating disorders before and during pregnancy might be associated with both binge drinking and macrosomia, which might impact the results of the association between binge drinking and macrosomia.

Reviewer #2: This is a retrospective cohort study on whether maternal alcohol drinking status pre-conception is associated with an increased risk of fetal macrosomia, and whether binge-alcohol drinking pre-conception may be an independent risk factor for fetal macrosomia. I have a few comments for the authors:

Introduction:

- This section references reports from the CDC and other studies that appear to be focused on US women, however there are also comments on alcohol consumption in different countries. Recommend that authors are clear what country/population they are citing as it is important to report the statistics with that additional context.

- Authors commented that the impact of ethanol consumption pre-conception on fetal development and postnatal growth is unclear. Recommend that it is important to highlight that the first trimester of pregnancy is the most susceptible to teratogens and that a large % of women are unaware they are pregnant until 4-6 weeks gestation.

Methods:

- It is important to clarify that diseases excluded are "pre-gestational" diabetes mellitus - so to not confuse with gestational diabetes mellitus.

- Unclear why only hepatitis A and B are excluded, but not hepatitis C or other underlying causes of liver-failure unrelated to alcohol consumption

- Smoking is listed as a "traditional risk factor" for macrosomia, however the literature on smoking suggests it is associated with fetal growth restriction and preterm birth

- If smoking is highlighted, then polysubstance use would be appropriate to note for all the participants too

- Need to define the acronym GDM - gestational diabetes - before using it and also clarify how this was diagnosed (eg. 1hr or 2hr glucose tolerance test, HbA1c?)

- Need to explain how congenital anomalies were detected, is this based on in-utero ultrasound or routine fetal anatomic survey, or postnatal assessment

Results:

- Page 12, Instead of referencing number of drinks = "cups" - recommend authors use a volume (e.g. drinks - 8oz)

Discussion:

- Authors reference a former rat study regarding fetal macosomia/metabolics, however there has been a recent non-human primate study of alcohol consumption (Lo et al. AJOG 2021) that noted no difference in fetal weight between controls and alcohol-exposed. Recommend citing other more recent and translational animal studies.

Table 1:

- It would be interesting to also include maternal race/ethnicity as well as other polysubstance use besides smoking

- Recommend using the words "vaginal delivery" rather than "normal delivery"

Table 2:

- Would be more relevant to include maternal HbA1c values rather than a random glucose as part of the comprehensive metabolic panel

Table 3:

- Need to define criteria for gestational diabetes as well as pregnancy-induced hypertension in the text or table legend

Table 4:

- Recommend adding 1 and 5 min apgar scores to this table

- Recommend clarifying the admission is to the "neonatal intensive care unit"

- Instead of writing "boys" or "girls" recommend using "male" and "female"

6. PLOS authors have the option to publish the peer review history of their article (what does this mean?). If published, this will include your full peer review and any attached files.

Reviewer #1: No

Reviewer #2: No

---

## [Author Response · Author response to Decision Letter 0]

13 May 2022

Editor Board

PLOS ONE

Dear Editors,

I would like to re

submit a revised version of the manuscript for publication in PLOS One , titled ‘Binge

alcohol drinking before pregnancy is closely associated with the development of macrosomia: Korean

pregnancy registry cohort’. The manuscript ID is PONE D 21 31508.

We appreciate you giving us the opportunity to revise our manuscript. We have carefully considered each

of the edit or’s and reviewers’ comments and suggestions, which helped us improve our manuscript. We

have provided point by point responses to all the reviewers’ comments in the attached letter.

We hope that our revised manuscript is now suitable for publication in

P LOS One . We look forward to

hearing from you.

Sincerely yours,

Won

Ho Kim, Ph.D.

Director

Division of Cardiovascular Disease Research,

Department of Chronic Disease Convergence Research, National Institute of Health,

187, Osong Saengmyeong 2

ro, Osong eu p, Cheongju city,

Chungbuk, Korea, 28159,

Tel: +82

43 719 8650; Fax: +82 43 719 8689; E mail: jhkwh@nih.go.kr

Please, check the attached file as " Response to reviewers".

---

## [Decision Letter · Decision Letter 1]

8 Jun 2022

PONE-D-21-31508R1Binge alcohol drinking before pregnancy is closely associated with the development of macrosomia: Korean pregnancy registry cohortPLOS ONE

Dear Dr. (KNIH),

Thank you for submitting your manuscript to PLOS ONE. After careful consideration, we feel that it has merit but does not fully meet PLOS ONE’s publication criteria as it currently stands. Therefore, we invite you to submit a revised version of the manuscript that addresses the points raised during the review process.

The reviewers have identified a couple of minor points that should be addressed.==============================

We look forward to receiving your revised manuscript.

Kind regards,

Emily W. Harville

Academic Editor

PLOS ONE

Journal Requirements:

Reviewers' comments:

Reviewer's Responses to Questions

**Comments to the Author**

1. If the authors have adequately addressed your comments raised in a previous round of review and you feel that this manuscript is now acceptable for publication, you may indicate that here to bypass the “Comments to the Author” section, enter your conflict of interest statement in the “Confidential to Editor” section, and submit your "Accept" recommendation.

Reviewer #1: (No Response)

Reviewer #2: (No Response)

2. Is the manuscript technically sound, and do the data support the conclusions?

Reviewer #1: Partly

Reviewer #2: Yes

3. Has the statistical analysis been performed appropriately and rigorously? 

Reviewer #1: Yes

Reviewer #2: I Don't Know

4. Have the authors made all data underlying the findings in their manuscript fully available?

Reviewer #1: Yes

Reviewer #2: Yes

5. Is the manuscript presented in an intelligible fashion and written in standard English?

Reviewer #1: Yes

Reviewer #2: No

6. Review Comments to the Author

Reviewer #1: 1. Why was the lifetime alcohol drinking pattern instead of the past year or 6 months drinking pattern selected? The past year or 6 months drinking pattern might be more relevant to pregnancy. I am asking because alcohol drinking patterns can vary a lot in different age groups (e.g., 20s vs. 30s).

2. I might have misunderstood. But why “Former or current” smokers were low-risk and never smokers were high-risk?. Also, I saw how you categorized different levels of physical activity in the footnote of Table 1, but not in the main text. This should be added to the main text as well.

Reviewer #2: Recommend that the authors consider the following:

- As there are no women with hepatitis C in the pregnancy registry, it is more comprehensive to say that patients with Hepatitis A, B and C were excluded from the study. Rather than just Hepatitis A and B alone.

- Even though the patient population was all Korean and that's why race/ethnicity was not reported, it is important to state this in the results section as part of the demographics. Otherwise, it will appear like the study did not further analyze by race.

7. PLOS authors have the option to publish the peer review history of their article (what does this mean?). If published, this will include your full peer review and any attached files.

Reviewer #1: No

Reviewer #2: No

---

## [Author Response · Author response to Decision Letter 1]

27 Jun 2022

"Response to reviewer" was attached. Please check it.

---

## [Editor Report · Decision Letter 2]

28 Jun 2022

Binge alcohol drinking before pregnancy is closely associated with the development of macrosomia: Korean pregnancy registry cohort

PONE-D-21-31508R2

Dear Dr. (KNIH),

We’re pleased to inform you that your manuscript has been judged scientifically suitable for publication and will be formally accepted for publication once it meets all outstanding technical requirements.

Kind regards,

Emily W. Harville

Academic Editor

PLOS ONE
---

## [Editor Report · Acceptance letter]

1 Jul 2022

PONE-D-21-31508R2 

Binge alcohol drinking before pregnancy is closely associated with the development of macrosomia: Korean pregnancy registry cohort 

Dear Dr. (KNIH):

I'm pleased to inform you that your manuscript has been deemed suitable for publication in PLOS ONE. Congratulations! Your manuscript is now with our production department. 

Kind regards, 

on behalf of

Dr. Emily W. Harville 

Academic Editor

PLOS ONE